# Expression, Purification, and Characterization of Anti-*Zika virus* Envelope Protein: Polyclonal and Chicken-Derived Single Chain Variable Fragment Antibodies

**DOI:** 10.3390/ijms21020492

**Published:** 2020-01-13

**Authors:** Pharaoh Fellow Mwale, Chi-Hsin Lee, Liang-Tzung Lin, Sy-Jye Leu, Yun-Ju Huang, Liao-Chun Chiang, Yan-Chiao Mao, Yi-Yuan Yang

**Affiliations:** 1Ph.D. Program in Medical Biotechnology, College of Medical Science and Technology, Taipei Medical University, Taipei 11031, Taiwan; pmwale@medcol.mw (P.F.M.); chihsine@msn.com (C.-H.L.); 2School of Medical Laboratory Science and Biotechnology, College of Medical Science and Technology, Taipei Medical University, Taipei 11031, Taiwan; jonsauwi@tmu.edu.tw; 3Graduate Institute of Medical Sciences, College of Medicine, Taipei Medical University, Taipei 11031, Taiwan; ltlin@tmu.edu.tw (L.-T.L.); cmbsycl@tmu.edu.tw (S.-J.L.); 4Department of Microbiology and Immunology, School of Medicine, College of Medicine, Taipei Medical University, Taipei 11031, Taiwan; 5College of Life Sciences, National Tsing Hua University, Hsinchu 30013, Taiwan; axe956956@gmail.com; 6Division of Clinical Toxicology, Department of Emergency Medicine, Taichung Veterans General Hospital, Taichung 40705, Taiwan; doc1385e@gmail.com; 7Core Laboratory of Antibody Generation and Research, Taipei Medical University, Taipei 11031, Taiwan

**Keywords:** *Zika virus* (ZIKV), envelope protein, phage display technology, single chain variable fragments (scFv), immunoglobulin yolk (IgY), chicken immunization

## Abstract

*Zika virus* (ZIKV) is a new and emerging virus that has caused outbreaks worldwide. The virus has been linked to congenital neurological malformations in neonates and Guillain–Barré syndrome in adults. Currently there are no effective vaccines available. As a result, there is a great need for ZIKV treatment. In this study, we developed single chain variable fragment (scFv) antibodies that target the ZIKV envelope protein using phage display technology. We first induced an immune response in white leghorn laying hens against the ZIKV envelope (E) protein. Chickens were immunized and polyclonal immunoglobulin yolk (IgY) antibodies were extracted from egg yolks. A high-level titer of anti-ZIKV_E IgY antibodies was detected using enzyme-linked immunosorbent assay (ELISA) after the third immunization. The titer persisted for at least 9 weeks. We constructed two antibody libraries that contained 5.3 × 10^6^ and 4.5 × 10^6^ transformants. After biopanning, an ELISA phage assay confirmed the enrichment of specific clones. We randomly selected 26 clones that expressed ZIKV scFv antibodies and classified them into two groups, short-linker and long-linker. Of these, four showed specific binding activities toward ZIKV_E proteins. These data suggest that the polyclonal and monoclonal scFv antibodies have the diagnostic or therapeutic potential for ZIKV.

## 1. Introduction

*Zika virus* (ZIKV) is an arthropod-borne virus and a member of the *Flavivirus* genus belonging to the *Flaviviradae* family. ZIKV is transmitted by *Aedes aegypti* and *Aedes albopictus* mosquitoes, as well as vertical transmission from mother to fetus and direct blood or biological fluids contact via blood transfusion or sexual contact [1,2,3,4]. The rapid spread of ZIKV is a significant threat to the human population [5]. ZIKV has been associated with several health issues, such as meningoencephalitis and myelitis in adults [6]; autoimmune disorders such as Guillain–Barré syndrome (GBS) in adults [7,8]; and microcephaly, which occurs in fetuses who were born from infected pregnant women [9]. Following its outbreak in Brazil, scientists reported an association between ZIKV and microcephaly cases in infants [9]. Since its discovery in Uganda in 1947 [10], the virus has caused outbreaks in many countries, such as Micronesia in 2007 and Brazil in 2015. Research has also suggested that individuals with compromised immunity could be more susceptible to ZIKV infection and disease development [11]. Much effort has been made towards preventing and curing ZIKV infections. However, no vaccines or drugs are currently approved on the market for the disease-causing ZIKV.

The ZIKV is an enveloped, positive single-stranded RNA virus that contains approximately 10,800 nucleotides and is closely related to the dengue virus (DENV), yellow fever virus (YFV), and West Nile virus (WNV) [12,13,14]. The ZIKV genome encodes three structural proteins and seven non-structural proteins. The structural proteins form viral particles, whereas non-structural proteins are involved in viral replication, viral assembly, and evasion of the host immune response. The envelope (E) protein forms a dimer on the smooth surface on the mature virus [15,16,17], and participates in receptor binding and fusion to the host. ZIKV infection is generally misdiagnosed because its signs and symptoms overlap with other endemic viral infection such as dengue and chikungunya [18,19]. Thus diagnosis of asymptomatic ZIKV infection relies heavily on serological evidence rather than clinical presentation [19]. The outbreaks of ZIKV in different countries emphasize the importance of establishing new and rapid diagnostic methods as well as development of an effective therapy that prevents its spread and infection.

Passive immunization using phage display technology has been used for the selection of single-chain fragment (scFv), the monoclonal antibodies with high specificity to the target of the interest, from antibody libraries [20] and, as a result, can potentially be applied as a diagnostic tool or therapeutic agent. Furthermore, scFv has a short half-life, such that if applied as treatment it can easily be eliminated from the body system. In phage display technology, the complementary deoxyribonucleic acid (cDNA) is employed as a template to amplify the variable domains of the heavy- and light-chain genes [21] of scFv. cDNA, using reverse transcription polymerase chain reaction (RT-PCR) [22], was converted from a total RNA of the spleen of immunized chicken. The phage display is a cost-effective method compared to hybridoma [23]. The latter technology requires expensive materials and is difficult to operate, although the former allows rapid generation time and selection of antibodies against unlimited antigens, biological or not [24]. On the other hand, studies have shown that passive immunization is a therapeutically potent method against viral infectious diseases. The chicken immunoglobulin yolk (IgY) antibody is a potential alternative for human immunoglobulin gamma (IgG) antibody immunotherapy [25]. IgY is an antibody that has several advantages, such as easy production from locally available chickens and undemanding extraction from purified egg yolk. The IgY antibody is also tolerable to low pH, making it a potential candidate for oral administration. Most importantly, it has a long shelf life, making it an appropriate alternative for vaccine development [26].

In this study, we report the first analysis of the immune response in chicken through determination of the polyclonal immunoglobulin yolk (IgY) against truncated *Zika virus* envelope (ZIKV_E) protein of the Brazilian lineage ZIKV strain [27]. We also demonstrate the specificity of the chicken anti-ZIKV_E single chain variable fragment (scFv) antibodies against the truncated ZIKV_E recombinant proteins. This ZIKV strain has potential for many outbreaks worldwide [27]. Together, our experimental systems show potential as a diagnostic agent for the *Zika virus* envelope protein and may be developed into a ZIKV_E vaccine.

## 2. Results

### 2.1. Expression, Purification, Immunization, and Characterization of ZIKV_E Recombinant Proteins

The ZIKV antigen was prepared for chicken immunization. The fragments of the ZIKV_E genes were amplified by PCR using designed primers (Appendix A and Table 1). Fragments were then cloned into the pET-21a plasmid vector, and expressed as His-tagged ZIKV_E recombinant proteins (Figure 1A). BL21 (DE3) *Escherichia coli* cells were used to express isopropyl-β-D-thiogalactoside (IPTG)-induced gene fragments. We managed to express ZIKV_E F2 and ZIKV_E F3 recombinant proteins successfully with an expected molecular mass of 27 and 17 kDa (Figure 1B), respectively. The proteins on SDS-PAGE were stained with Coomassie blue dye. Next, we performed the western blot assay. The ZIKV_E recombinant proteins were transferred to a polyvinylidene difluoride (PDVF) membrane and detected using mouse anti-His tag antibodies (1/5000 dilutions) to confirm the ZIKV_E recombinant protein expression (Figure 1C).

The antibody production in chickens was observed through immunization of His-fused recombinant proteins. The chicken received an intramuscular injection on a thigh of 50 micrograms (µg) mixture of the purified ZIKV_E F2 and ZIKV_E F3 proteins each 7 days for 56 days. A total of 50 µg of two ZIKV_E proteins was resuspended in filtered phosphate buffered saline (PBS) plus Freund’s adjuvant (Sigma-Aldrich, Inc., St Louis, MO, USA). The first shot was mixed with 11 µL of ZIKV_E F2 (34.705 µg), 4.7 µL of the ZIKV_E F3 (15.0447 µg), 484.3 µL of the filtered PBS, and then topped up with 500 µL of complete Freund’s adjuvant to make a mixture of 1 milliliter (mL). The immunization schedule was designed to run for 56 days (Figure 1D). The eggs were collected before and after immunization each week for purification (Figure 1D).

### 2.2. Purification and Characterization of Polyclonal Anti-ZIKV_E IgY Antibodies

The polyclonal IgY antibodies were purified from the collected eggs by precipitation using saturated sodium sulfate solution with minimal modification [28], and examined by SDS-PAGE. The reduced immunoglobulin yolk (IgY) antibody (using 2-mercaptoethanol) contained a major band of about 65 kDa (heavy chain) and a minor band of about 26 kDa (light chain) (Figure 2A). The average concentration of purified IgY antibodies were estimated to be approximately 9.75 mg/mL at each time point using bicinchoninic acid (BCA) protein quantification assay. To evaluate polyclonal IgY antibody titer, we tested immunized chickens for increased humoral immune response during the 7 weeks period of immunization. We performed an enzyme-linked immunosorbent assay (ELISA). The analysis showed strong humoral immune responses in the immunized chickens (Figure 2B). The results in Figure 2B indicated that the elicited immune response in chickens started increasing after the first shot (first immunization) against ZIKV_E F2 proteins, then steadily increased until reaching a plateau with the third immunization for the ZIKV_E F2 protein (Figure 2B), although immune response was rapidly induced for ZIKV_E F3 protein after first immunization and then reached a plateau after the second immunization (Figure 2B).

We monitored the antibody response in chickens by using immunoglobulin yolk (IgY) antibodies and compared the pre-immunization through the last immunization (seventh immunization) titers (Figure 2C–J). The IgY antibodies of the seventh immunization had a significant binding activity (optical density (OD) > 1.0) at a dilution of 1/256,000 (Figure 2J), compared to the binding activity of pre-immunization, which showed an OD = 1.0 at a dilution of 1/500 (Figure 2C). The binding activities against BSA showed no significant reaction. Together, the data suggest that the immune response was effectively raised at the third immunization for ZIKV_E F2 protein, whereas for ZIKV_E F3 protein was raised at the second immunization.

### 2.3. Bio-Panning Process and Characterization of Phage-Displayed Chicken Anti-ZIKV_E scFv Antibody Libraries

Next, we prepared anti-ZIKV_E antibody libraries. Briefly, the complementary deoxyribonucleic acid (cDNA) was synthesized using extracted total RNA from chicken spleens and was used to amplify the variable regions of heavy chain (VH) and light chain (VL). The amplified VH and VL gene products were joined by a short- or a long-linker to make the full-length of single-chain variable fragment (scFv) gene, and it was cloned into a pComb3XSS vector. After cloning the scFv gene, two antibody libraries were constructed, scFv-S and scFv-L, containing 5.3 × 10^6^ and 4.5 × 10^6^ transformants, respectively.

After infection by VCSM13 helper phages, the phages displaying scFv antibodies were used in the bio-panning steps. The phage eluted titers were measured as approximately 10^3^ colony-forming units (cfu) in the first cycle against ZIKV_E F2 protein (Figure 3A). The titer increased by 1000 cfu in the second cycle, the pattern remained the same in third round, and it steadily increased to 10^6^ cfu in the fourth cycle (Figure 3A). The phages displaying scFv with long-linker indicated a titer at 10^4^ cfu in the first cycle, a small increase in the second, and then the titer steadily increased to 10^7^ cfu in the third and fourth cycles (Figure 3A). On the other hand, the eluted phages were also tested against ZIKV_E protein. The phage containing scFv with short-linker showed 10^4^ cfu, and then steadily increased above 10^6^ cfu, whereas the phage containing long-linker started with 10^4^, then decreased a small amount in second cycle, and steadily increased to 10^7^ cfu in the fourth cycle (Figure 3B). A similar pattern was previously observed in our published work [29], which suggested that specific phage binders were successfully enriched from the start until the end of the bio-panning procedure. We further confirmed the results of the bio-panning process using ELISA. The amplified phages were added to the ELISA plate with wells coated with purified ZIKV_E recombinant proteins. The original phage antibody libraries were compared with phages amplified in the first through fourth rounds. The result indicated that the binding activity of harvested phages was significantly enhanced after the second panning in long-linker against ZIKV_E F2 proteins (Figure 3C). Those against ZIKV_E F3 proteins showed a steady increase up to the fourth round with the short-linker, whereas the long-linker produced sharp increase after first round and then plateaued from the second round onwards (Figure 3D).

### 2.4. Sequence Analysis, Expression, Purification, and Characterization of Anti-ZIKV_E scFv Antibodies

We first tested the binding activities of the antibody after biopanning. A total of 39 colonies were randomly picked from the Luria-Bertani (LB) agar plate after the fourth round of antibody selection and cultured for DNA extraction. The DNA extracted was transformed into competent *Escherichia coli* (TOP10F’ strain) to express proteins. ELISA assay was performed to determine the binding. The results showed 12 clones had strong binding affinity against ZIKV_E F2 protein (Figure 4A), 7 clones against ZIKV_E F3 protein (Figure 4B), and no binding affinity of anti-ZIKV_E with long-linker against ZIKV_E F3 protein (Figure 4C). Two controls were used as negative and positive (ZIKVEIgY). The antibodies had no binding affinity to BSA (Figure 4A–C). The potential positive antibody clones were grouped according to similarities and differences of nucleotide sequence, only three clones were different and the rest were the same (data not shown). Therefore, four scFv clones were selected. These four clones from two libraries were analyzed in terms of their VL and VH nucleotide sequences and compared to the chicken germ line. The overall mutation rate of amino acids in complementarity determining regions (CDRs) was between 20% to 100% and 0% to 19% in framework regions (FRs) (Table 2 and Figure 4D). The higher mutation rate corresponds to improved affinity, whereas a lower mutation rate exhibits low affinity [30]. The CDR regions demonstrated a similar phenomenon, which proved good binding affinity to ZIKV_E recombinant proteins. On the other hand, the greater variation in the CDRs compared with the FR regions implied that the scFv clones were selected from immunized chicken B cells, not directly from naïve B cells. This is evidence of an antigen-driven antibody response after immunization. Together, the data suggest that two libraries contain scFv antibodies with high binding activities. The expressions of scFv antibodies were also analyzed. After overnight incubation and isopropyl-β-D-thiogalactoside (IPTG) induction, the anti-ZIKV_E scFv antibodies were purified and analyzed by SDS-PAGE and western blots. The results showed expected bands of molecular masses ranging from 25 to 30 kilodaltons (kDa) (Figure 4E). This suggested that the scFv antibodies were expressed and purified successfully. The purified scFv antibodies were confirmed, which was detected by mouse anti-human influenza virus hemagglutinin (HA) tag antibodies, followed by rabbit anti-mouse immunoglobulin gamma (IgG) horseradish peroxidase (HRP)-conjugated antibodies (Figure 4F).

### 2.5. Characterization of Binding Specificity of Anti-ZIKV_E scFv and Polyclonal Anti-ZIKV_E IgY Antibodies

The ZIKV_E proteins were further analyzed using Coomassie blue-stained SDS-PAGE (Figure 5A). Mouse anti-His tag detection on western blot membrane confirmed the protein production (data not shown). The molecular masses of the two different ZIKV_E truncated proteins were ≈27 or ≈17 kDa, respectively (Figure 5A). No reactivity was detected against bovine serum albumin (BSA). ZIKVEF2L01 scFv, ZIKVEF3S06 scFv, ZIKVEF3S07 scFv, and ZIKVEF3S13 scFv recognized ZIKV_E truncated proteins in western blot (Figure 5A); the same pattern of binding activities was observed in ELISA assay (Figure 5B), especially ZIKVEF2L01 scFv. However, two scFvs (ZIKVEF3S06 and ZIKVEF3S13) showed different patterns of binding activities. This result suggested that the epitopes of these two scFvs antibodies on ZIKV_E F2 protein might have been exposed during the SDS-gel electrophoresis process, compared to native form protein in ELISA assay. A non-reactivity pattern of ZIKVEF3S07 scFv to ZIKV_E F2 protein was also observed both from western blot and ELISA assay (Figure 5A,B). ZIKVEIgY and anti-OmpAFLS11 scFv antibodies were used as positive and negative controls, respectively (Figure 5A,B).

Furthermore, the purified scFv antibodies were used to test the ZIKV_E-truncated proteins coated on the ELISA wells plate. The results showed that the ZIKVEF2L01 scFv had lower binding activity (optical density less than 1.0) against ZIKV_E F3 protein but moderate reactivity against ZIKV_E F2 protein (optical density greater than 1.0), whereas ZIKVEF3S06 scFv, ZIKVEF3S07 scFv, and ZIKVEF3S13 scFv showed moderate binding activities (optical density greater than 1.0) to the same ZIKV_E F3 protein (Figure 5B). Meanwhile, these three scFv (ZIKVEF3S06, ZIKVEF3S07, and ZIKVEF3S13) antibodies showed no reactivity against ZIKV_E F2 and BSA proteins in ELISA assay (Figure 5B). Taken together, these data implied that the specific immune response was induced against ZIKV_E F2- and ZIKV_E F3-truncated proteins in chickens.

The binding ability of scFv antibodies were further tested using flow cytometry. The result showed a shift of a histogram to the right, which meant that ZIKVEF2L01 scFv and ZIKVEF3S06 scFv recognized ZIKV_E protein in a transfected Vero cell (Figure 5C). However, the remaining scFv (ZIKVEF3S07 and ZIKVEF3S13) could not recognize the ZIKV_E protein expressed by monkey kidney cells (Figure 5C). The 4G2 anti-flavivirus group antibody (commercial antibody) was used as a positive control (Figure 5C). Thus, these data implied that the immune response was induced for both ZIKV_E F2 and ZIKV_E F3 proteins in chickens.

### 2.6. Competitive ELISA Assay of Anti-ZIKV_E scFv Antibodies

The assay further confirmed the binding abilities of anti-ZIKV_E scFv antibodies. First, the checkerboard titration was performed to determine the appropriate concentration for use in the subsequent assay (Appendix A). After antibody titration, 20 µg/mL was chosen as the concentration to use. Each scFv antibody was incubated with the free form of the ZIKV_E recombinant protein separately, and 1 h later it was added to the ELISA wells plate. The percentage of inhibition was calculated using optical densities in the absence of the free form of ZIKV_E recombinant proteins divided by the presence of serially diluted concentration of the free form of ZIKV_E recombinant proteins. The dissociation constant (*K_D_*) of each scFv was calculated using the method of the Klotz plot (Table 3). As indicated, the binding abilities of these scFvs were gradually decreased in a dose-dependent manner (Figure 6). The results showed that the K_D_ values of ZIKV_E scFv were 1.49 × 10^−5^ to 2.04 × 10^−8^ M, which suggest that ZIKVEF2L01 scFv had the strongest affinity (2.04 × 10^−8^ M) whereas ZIKVEF3S07 scFv had the weaker affinity (1.49 × 10^−5^ M) to the ZIKV_E protein. Nevertheless, these results suggest that ZIKV_E scFv antibodies had the binding affinity for recombinant ZIKV_E-truncated proteins.

## 3. Discussion

ZIKV has caused a number of outbreaks since its discovery in the Zika forest of Uganda in 1947, and it continues to pose a major public health threat [31]; in addition, it can be misdiagnosed with other viruses of similar family lineages, such as dengue and/or chikungunya viruses [32]. As such, antibodies can be a valuable therapeutic and diagnostic tool [32]. Both polyclonal and monoclonal antibodies may be good options in this regard. The polyclonal immunoglobulin yolk (IgY) antibodies produced in chicken laying eggs have various advantages, such as high affinity for detection, multiple epitope recognition [33], low cost, and ease in purification compared to blood collection [34]. On the other hand, monoclonal-derived antibodies are also good tools for detection because they demonstrate specific binding affinity to their targets (ZIKV_E proteins).

The characterization and isolation of the ZIKV envelope protein has been described for *Zika virus* infection in humans, primates, and other species such as rhesus monkeys and mice [35]. However, an effective antibody response has not yet been characterized. Nevertheless, the envelope protein can elicit humoral and cell-mediated immune response [36,37,38].

In this paper, we designed the sketch diagram of ZIKV_E protein (Figure 1A), then analyzed and confirmed the binding activities of four anti-ZIKV_E scFvs (ZIKVEF2L01, ZIKVEF3S06, ZIKVEF3S07, and ZIKVEF3S13). These scFvs exhibited solubility when expressed in a bacterial system. ZIKVEF3S06 and ZIKVEF3S07 were the least soluble, with half of the expressed product remaining inside the bacterial cell cytoplasm and other half being purified for analysis in respective assays. In order to achieve a sufficient production, a large volume was cultured. The purified scFvs exhibited a good binding ability against the recombinant ZIKV viral envelope antigen in BL-21 *Escherichia coli* (Figure 1B,C). Confirmation of the immunogen protein expression was confirmed using SDS-PAGE and western blot (Figure 1B,C). As a result, the anti-ZIKV_E in this study is a promising antibody for use as an in vitro diagnostic for ZIKV infection.

After the recombinant ZIKV_E protein chicken immunization (Figure 1D), IgY antibody proteins were purified from immunized eggs (Figure 2A). The purity of the antibody proteins was determined using SDS (Figure 2A). Importantly, the titer showed a humoral immune responses was elicited in the chicken. The result showed a steady increase in anti-ZIKV_E IgY antibodies from the second week post-immunization and reached a plateau after the third week immunization (Figure 2B). Similarly, other studies using different immunization antigens in egg-laying chickens showed a similar pattern [39]. The specificity of IgY antibodies to the ZIKV_E proteins was further confirmed using ELISA. It was determined that the titer of IgY antibodies showed strong binding to the target (antigen) when compared to non-immunized IgY antibodies (Figure 2B–J). The results displayed strong immune reactivity against recombinant ZIKV_E-truncated proteins with IgY antibodies from first immunization through seventh immunization compared with the pre-immunization (Figure 2B–J). Other studies have shown a similar outcome being observed with anti-hepatitis A virus immunoglobulin Y purified from hens’ eggs [40]. The results indicated that humoral immune response in chicken was induced. Together, this implies that the recombinant proteins can be used as a vaccine to boost immunity for preventing ZIKV infection.

A phage display technique for scFv antibodies was used in this study against purified viral envelope antigen. This technique is widely applied for antibody selection against an antigen target [41]. The phage display system requires constructed antibody libraries from which the specific antibodies are selected [42]. Following this observation, we constructed two libraries containing 5.4 × 10^6^ and 6.5 × 10^6^ plaque-forming units (pfu) from seventh immunized chicken. Figure 3A–D showed an increase of phage-expressing scFvs at the fourth round of bio-panning. Taken together, these data suggest that anti-ZIKV_E scFv antibodies were effectively enriched at this last cycle of antibody selection.

We analyzed the randomly selected scFvs and their gene sequences, and then compared them to the chicken germline sequence. A total of 19 scFv clones displayed good binding activities against the ZIKV_E-truncated antigens after selection (Figure 4A,B). However, the anti-ZIKV_E scFv with long-linker against ZIKV_E F3 protein did not have binding activities to ZIKV_E proteins (Figure 4C). Nevertheless, we managed to select four potential scFv antibodies with specific binding to ZIKV_E proteins that can be used in the detection of recombinant protein of *Zika virus* envelope in western blot, ELISA, and flow cytometry assays. After selection, the scFv gene sequences were also determined. The complementarity determining regions (CDRs) and the framework regions (FRs) showed different amino acid residues, which denote mutations. The CDR regions were responsible for the immunogen binding activities, which were revealed by mutation rates. The scFvs showed 20–100% and 0–19% mutation rates, respectively, in both light- and heavy-chains when compared to the chicken germline (Figure 4D and Table 2). These mutation rates indicated that anti-ZIKV_E scFvs antibodies elicited an immunogenic-driven chicken humoral immune response rather than a naturally induced naïve antibody response. Expression and purification levels in TOP10F’ *Escherichia coli* were analyzed using SDS-PAGE in order to determine their molecular weight in kilodaltons (Figure 4E); western blot analysis was used to confirm the expression of the ZIKV_E scFvs antibody proteins as shown in Figure 4F. According to the results, scFvs proteins were successfully expressed and purified.

The anti-ZIKV_E scFvs antibodies synthesized in this study showed positive binding against viral envelope proteins. Previous studies conducted scFvs binding activity analysis against different viral antigens including envelope proteins of the viruses [43]. The results indicated that selected scFv fragments recognized viral envelope proteins, which are the virus integral proteins for attachment and infection to the host cells. This implies that strong binding activities can be achieved. We observed binding activities of scFv antibodies against ZIKV_E proteins (Figure 5A). Furthermore, the ZIKVEF2L01, ZIKVEF3S06, and ZIKVEF3S13 scFv showed antigen recognition against both truncated proteins (fragments 2 and 3), whereas ZIKVEF3S07 scFv recognized fragment 3 protein only (Figure 5A). This implies that the latter scFv antibody has a unique binding epitope to the truncated ZIKV_E protein. The same pattern of reactivity against the viral envelope antigen through ELISA, with optical density (OD) greater than 0.5 nanometer (nm), was observed. No reactivity was depicted when probed with OmpAFLS11 scFv as a negative control (Figure 5B). However, higher reactivity was achieved with anti-ZIKV_E IgY antibody with the OD > 2.0 nm as positive control (Figure 5B). The binding activities were also showed through flow cytometry analysis. In the flow data, two scFv antibodies showed good binding activities, whereas the other two (ZIKVEF3S07 scFv and ZIKVEF3S13 scFv) did not display a shift of histogram to the right, which meant the antibodies were not bound to the antigen (ZIKV_E proteins) (Figure 5C). The anti-ZIKV_E scFv antibodies were further characterized through competitive binding assay (Figure 6). The results demonstrated the inhibition of antigen by the anti-ZIKV_E scFv antibodies. This suggests that if the antibodies can be used in animal studies, they will show good neutralization effects of *Zika virus*, thereby preventing infection.

In this study, ZIKV_E immunoglobulin yolk (IgY) antibodies were extracted from the egg yolk of chicken-laying eggs immunized with a truncated recombinant E protein of *Zika virus*. We described the binding activity of anti-ZIKV_E IgY antibodies for recombinant ZIKV_E proteins and validated ZIKV detection in vitro. The in vivo neutralization of *Zika virus* infection using the IgY and scFv antibodies were not performed due to the restrictive policy of handling live or pseudo particles such as the *Zika virus*. In addition, we developed specific anti-ZIKV_E scFv antibodies using phage display antibody technology. Although we observed effective binding towards the target antigen in in vitro assays, further validation of the neutralization activities of these polyclonal and monoclonal antibodies through an in vivo model would further elucidate the function of these antibodies as a viral therapeutic.

## 4. Materials and Methods

### 4.1. Plasmid Vectors, Bacterial Strains, Media, Cell Lines, and Antibodies

We used pComb3XSS (phagemid vector) [44] and pET-21a vector (Merck Millipore, Darmstadt, Germany) plasmid vectors, and TOP10F’ *E. coli* (Thermo Fischer Scientific, Massachusetts, NH, USA), BL-21(DE3) *E. coli* (New England BioLabs (NEB), Massachusetts, USA), and ER2738 *E. coli* (NEB, Boston, MA, USA) bacterial strains. Luria-Bertani (LB) medium (both broth and agar; 10 g bio-tryptone, 5 g yeast extract, 10 g sodium chloride, pH 7.4) and super broth (SB; 30 g bio-tryptone, 20 g yeast extract, 10 g 3-(N-Morpholino)propanesulfonic acids (MOPs), pH 7.4) comprising 1 L were used. The media powders, antibodies, and enzymes were purchased from BioShop Canada Inc., Burlington, and supplied by Bioman Science Co. Ltd, Protech Technology Enterprise Co. Ltd, Taigen Bioscience Corporation, Thermo fisher scientific Inc, branches in Taiwan, which were used throughout the project. The Vero cells (a gift from Dr. Liang-Tzung Lin, Taipei Medical University) were used to express the ZIKV_E protein to mimic animal cell-expressing model. Secondary and primary antibodies such as horseradish peroxidase (HRP)-conjugated rabbit anti-mouse IgG and HRP-conjugated donkey anti-chicken IgY were sourced from Jackson ImmunoResearch Laboratories.

### 4.2. Preparation of ZIKV_E Gene Fragments

ZIKV_E recombinant protein was prepared by subcloning two overlapping fragment genes from the *Zika virus* envelope gene (strain Brazil-ZKV2015; GenBank#: KU497555) [45] cloned in pCDNA3.1 (+) plasmid vector (a gift from Dr. Liang-Tzung Lin, Taipei Medical University). The primers used for amplifying the fragment genes were (pZIKV_EF2-FW: TAAAATGGATCCATGATCAGGTGCATAGGAGTCAGC, pZIKV_EF2-RV: TAAATTCTCGAGGGCATGTGCGTCCTTGAACTCTAC) and (pZIKV_EF3-FW: AATTAAGGATCCGACATTCCATTACCTTGGCACGCT, pZIKV_EF3-RV: TTAATTCTCGAGGATTACGGGGTTAGCGGTTATCAAC). Thereafter, the F2 and F3 fragment genes were subcloned into His-tag containing pET-21a (+) plasmid vector (Novagen, Darmstadt, Germany) using *Bacillus amyloliquefaciens strain H* (*BamHI)* and *Xanthomonas holcicola* (*XhoI)* restriction enzymes and expressed in BL21 (DE3) *Escherichia coli*.

### 4.3. Expression and Purification of ZIKV_E Recombinant Proteins

We transformed the expression plasmid vector containing *ZIKV_E* gene into BL21 *Escherichia coli* (DE3) for expression of the recombinant proteins. An aliquot of 100 µL of *E. coli* was diluted into 5 mL of Luria-Bertani medium and culture for 2–3 h at 37 °C. When the bacteria reached a density (OD_600_) of greater than 0.4 but less than and/or equal to 0.8 at 37 °C, the isopropyl-β-d-1-thiogalactoside (IPTG) was added to a final concentration of 0.5 mM and the cells were grown overnight to induce the production of the recombinant proteins. Following expression, ZIKV_E proteins were purified using nickel beads (Sepharose), quantified with NanoVue machine (CoWin Biosciences, Beijing, China), and then stored at −20 °C for subsequent use.

### 4.4. Immunization of White Leghorn Chicken

We constructed the *Zika virus* envelope gene into a pET-21a vector, expressing and purifying it, which was then used to immunize white leghorn female chicken. After purification, a total of 50 micrograms (µg) of two fragments of ZIKV_E proteins was mixed with an equal volume of Freund’s complete adjuvant (Sigma-Aldrich, St. Louis, MO, USA) for the first intramuscular immunization on the thigh. ZIKV_E full-length and F4 fragments could be expressed, but could not be purified. A total of 300 µg of ZIKV_E recombinant proteins mixed with Freund’s incomplete adjuvants were used for the subsequent immunization at a period of 7 day intervals. The immunized chicken eggs were collected for 56 days before and after every 7 days in order to extract polyclonal immunoglobulin yolk (IgY) antibodies, as described in a previously published paper [46]. The non-immunized eggs were collected for use as the negative control.

### 4.5. Extraction of Anti-ZIKV_E IgY from Egg Yolk

The extraction of IgY from egg yolks was done according to the method described in a previous study [47]. In short, the white part was sieved from yolk by egg separator, 10 milliliters (mL) was collected, and the egg yolk was diluted with four volumes (40 mL) of Tris-buffered saline (TBS), pH 7.4. The mixture was vortexed thoroughly and incubated at room temperature (RT) for 30 min. After centrifugation (4500 revolutions per minute, rpm) for 20 min at RT, the supernatant was collected into a new 50 mL tube. Dextran sulfate solution was added per milliliter of supernatant, thoroughly mixed, incubated, and then was followed by the addition of calcium chloride per milliliter of supernatant, mixed again, and spun at the same number of revolutions. The collected supernatant was filtered through a ball of cotton wool. After adjusting the filtrate to 100 mL in TBS, 20 g of an anhydrous sodium sulfate was added slowly until it was completely dissolved and left to stand for 30 min at RT. Following incubation, centrifugation under the above conditions was performed, and the pellet was re-dissolved from precipitation in 10 mL TBS. After 20 min centrifugation, the supernatant was collected, and slowly 8 mL of 36% sodium sulfate solution was added, being left to stand for white precipitate to form. The pellet was re-dissolved in 5 mL TBS following centrifugation at RT for 20 min. To the final volume, 0.05% sodium azide was added for preservation and stored at −20 °C. Approximately 50 to 100 mg were obtained from one egg yolk in 5 mL. The IgY antibody concentration was determined by titration and was then analyzed on 12% denaturized sodium dodecyl sulfate-polyacrylamide gel electrophoresis (SDS-PAGE) to confirm its purity.

### 4.6. ELISA for Chicken Immune Response Analysis

Immunoglobulin yolk (IgY) antibodies were used to confirm the induction of humoral immune response in chicken. The analysis was performed after completion of immunization and eggs collection. The ELISA protocol was followed according to Engvall’s [48], Palmieri’s [49], and Coillie’s [50] method with minimal modifications. Briefly, ZIKV_E recombinant protein (10 µg) was coated in the 96 half-area well microplates (Becton Dickinson (BD), San Jose, CA, USA) and was incubated overnight at 4 °C. BSA protein was used as a negative control. The ELISA microplate wells were blocked with 200 µL of 5% skimmed milk for 1 h. After blocking, the plates were washed six times with PBS buffer containing 0.05% Tween-20 (Merck KGaA, Darmstadt, Germany). Following six more washing steps, 25 µL of primary antibody (anti-ZIKV-E IgY) diluted in 5% skimmed milk blocking buffer (1/3000 dilution) was added to each well, including control wells, and incubated for 1 h. After incubation, the wells were washed again (six times) and 25 µL of the donkey anti-chicken IgY HRP-conjugated antibody (Agrisera, Vannas, Sweden) was added in 1/10,000 dilution and incubated for 1 h. After six washes, 25 µl of the 3,3′,5,5′-tetramethylbenzidine (TMB; BD, San Diego, CA, USA) substrate was added to each well and incubated in the dark at room temperature for 13 min. The reaction was stopped by adding 13 µL of 1N HCl, which was evidenced by color change from blue to yellow. The absorbance values were read immediately at 450 nm using a plate reader (BioTek U.S, Northern Vermont, USA) and the optical densities were analyzed by Graph Pad Prism version 8.0 software (La Jolla, CA, USA).

### 4.7. Chicken Antibody Library Construction

The antibody libraries were constructed as described in the protocol of [51]. Briefly, the spleen cells obtained from sacrificed chicken were homogenized with iron beads to release the total RNA molecules. After the extraction, 10 μg of RNA as template was transcribed by reverse transcriptase (SuperScript RT kit, Thermo Fisher Scientific, England, United Kingdom) to synthesize the first-strand cDNA. This molecule (cDNA) was amplified with chicken specific primers to obtain gene products of heavy- and light-chain variable domains (VH and VL). Flexible 7- or 18-amino acid peptide linker connected the two domains (VH and VL) in order to synthesize the full-length single-chain variable fragment (scFv) through an overlap extension PCR. The resulting product was further cloned into a pComb3XSS, an scFv expression phagemid vector. The resultant plasmids DNAs were transformed into *Escherichia coli* strain TOP10F’ by electroporation (MicroPulser; Farmigdale, New York, NY, USA). Then, 10 µL of the transformants were plated on a Luria-Bertani agar plate (50 µg/mL ampicillin (Amp)) to calculate the library size. The rest of the transformed *E. coli* were added into 100 mL of super broth (SB) and then infected with VCSM13 helper phage (Stratagene, La Jolla, CA, USA). After spinning down to collect supernatant from bacterial pellet on the following day, the recombinant phages in the supernatant were precipitated with 4.48% polyethylene glycol 8000 (*w*/*v*) and 3.36% sodium chloride (*w*/*v*) [52] on ice for at least 30 min. After incubation and centrifugation, the phage pellets were re-suspended in phosphate-buffered saline (PBS) containing 1% bovine serum albumin (BSA) and 20% glycerol, and stored at −20 °C in a refrigerator.

### 4.8. Bio-Panning of Anti-ZIKV_E scFv from Chicken Antibody Library

In the panning protocol, the ZIKV_E recombinant protein (10 µg/mL) was coated on ELISA plates (Nunc, Roskilde, Denmark) and incubated overnight at 4 °C. After incubation, the well plate was blocked with 3% bovine serum albumin (BSA) (dissolved in 1×PBS) for 1 h at 37 °C, followed by washing thrice with 1×PBST (1×PBS + 0.05% Tween-20). A total of 50 µL of 10^12^ pfu (plaque forming unit) phage library was added and incubated at 37 °C for 2 h. After phage binding, the ELISA plate wells were washed 10 times at an interval of 30 times with vigorous pipetting up and down using filtered 1×PBST (1×PBS + 0.05% Tween-20). The bound phages were eluted using 100 µL of 0.2 M glycine-HCL elution buffer (reconstituted with 1 mg/mL BSA), pH 2.2, in each well. The eluate was immediately neutralized with 2 M Tris-base neutralization buffer. The eluted phages were amplified as described in a previously published article [53].

### 4.9. Anti-ZIKV_E scFv Antibodies Selection, Expression, and Purification after the Fourth Round of Bio-Panning

Bio-panning is a widely used method to select specific antibodies. In this study, the antibodies were selected against ZIKV_E recombinant proteins, and were bio-panned from phage library. The total DNA was extracted from the bacteria pellet following the high titer of phage binders in the last round (fourth) of bio-panning. The extracted total DNAs were transformed into a TOP10F *E. coli* (the competent cells) for the analysis of antibody protein expression. The transformed products were plated on a Luria-Bertani (LB) agar plate containing 50 µg/mL of ampicillin (Amp) and incubated at 37 °C. Individual colonies were randomly selected from the cultured plate, suspended in super broth (SB) media containing 1 M magnesium chloride (MgCl_2_) and 50 µg/mL of Amp, and cultured at 37 °C. After 8 h of incubation, 1 mM IPTG was added for overnight protein expression induction [46]. The cultured products were centrifuged at 3500 revolutions per minute (rpm) for 10 min to collect the bacterial cells pellet. The cells were harvested and lysed in His-binding buffer with urea containing 20 µM imidazole, 500 µM sodium chloride, 20 µM sodium phosphate, 6 M urea, pH 7.4, through sonication (Misonix, ultrasonic liquid processors, XL-2000 series). The supernatant consisted of anti-ZIKV_E scFv antibodies that were purified using nickel beads according to the manufacturer’s directions (GE Healthcare Bio-Sciences AB, Uppsala, Sweden).

### 4.10. SDS-PAGE and Western Blot Analysis

The purified ZIKV_E recombinant proteins were resolved by 15% SDS-PAGE under reducing conditions with beta-mercaptoethanol (β-me). For western blot analysis, rabbit anti-mouse IgG HRP-conjugated (10,000× dilution) (Jackson ImmunoResearch, West Grove, PA, USA) detected ZIKV_E recombinant proteins, and the blot was developed using the 3,3′-diaminobenzedine (DAB)/H_2_O_2_ system.

### 4.11. Enzyme-Linked Immunosorbent Assay (ELISA)

The binding ability of the phage-expressing anti-ZIKV_E scFv antibody was measured by ELISA. The 96 half-area well plates were coated with 10 µg/mL per 25 µL in each well of recombinant ZIKV_E proteins, and were incubated at 37 °C for 1 h. After incubation, the plate was blocked for 1 h at 37 °C with 5% skimmed milk dissolved in 1×PBS and washed six times with 1×PBST (1×PBS, 0.05% Tween-20). After 1 h blocking, the equal mixture of phage antibody with 5% skimmed milk 1×PBS was added at 25 µL per well and incubated at 37 °C for 2 h. Chicken anti-ZIKV_E IgY (seventh immunization) was used as a positive control, whereas BSA was used as a negative control. After incubation, the wells were washed six times with 1×PBST, followed by incubation with mouse anti-M13 HRP-conjugated antibody for 1 h at 37 °C. After washing six times, donkey anti-chicken IgY HRP-conjugated antibody (1/10,000 dilution) or rabbit anti-mouse IgG HRP-conjugated antibody (1/10,000 dilution) was added and incubated at 37 °C. After 1 h incubation, the wells were washed six times with 1×PBST, followed by development with 3,3′,5,5′-tetramethylbenzidine (TMB) (Sigma, Shanghai, China) for 20 min in the dark at room temperature. The reaction was stopped with 13 µL 1 N HCl solution in each well. The absorbances were read at 450 nm in a plate reader, and finally the data was analyzed with Graph Pad software version 8.0.

### 4.12. Expression of ZIKV_E Protein in Vero Cells

Vero cells were infected with the ZIKV particles (MOI = 1) for 72 h. At the end of the incubation, the cells were washed with cold 1×PBS, trypsinized, and fixed with 4% paraformaldehyde. After fixing and permeabilizing, the cells were then blocked with 3% BSA for hour at room temperature and incubated with 4G2 anti-flavivirus antibody (1:200) at 4 °C overnight. Cells were subsequently washed and incubated with the Alexa Fluor 488 secondary antibody at room temperature for 1 h. The cells were finally washed thrice with cold 1×PBS, resuspended in 1×PBS, and then measured with a flow cytometer. The data were analyzed by Flow Jo software.

### 4.13. Competitive ELISA

The assay was performed in order to determine the inhibition of anti-ZIKV_E scFv antibodies against ZIKV_E recombinant proteins. We mixed ZIKV_E proteins by double serial dilution (400 to 0.2 µg/mL) with an equal volume of individual anti-ZIKV_E scFvs (20 µg/mL) for 1 h at room temperature. After 1 h incubation, the mixtures were added to the wells that were coated with 10 µg/mL of ZIKV_E proteins and incubated for 1 h at 37 °C. Blocking and washing followed this step, as well as signal detection, as described in the ELISA assay steps above. After competitive ELISA results, the 50% inhibitory effect was tested. All the data were presented as the mean ± standard deviation from independent triplicated experiments. This brief protocol was consulted from a previously published article [54].

### 4.14. Sequence Analysis of Anti-ZIKV_E scFv Antibodies Genes

In order to confirm the identity of produced scFv antibodies in the bacteria system, the scFv DNA were extracted and their nucleotides were sequenced and compared with the chicken germ line. These nucleotide sequences of scFv-expressing clones were synthesized using the ompseq (5′-AAGACAGCTATCGCGATTGCAGTG-3′) primer through an autosequencer (ABI PRISM 377, Perkin-Elmer). Their putative amino acid sequence VL and VH genes were aligned with those of the chicken immunoglobulin germ line gene.

### 4.15. Statistical Analysis

The data were determined by GraphPad Prism version 8.0 software (La Jolla, CA, USA), which was calculated as the mean ± standard deviation (SD) for duplicated or triplicated absorbance reading of ELISA assay. The optical density reading (*p*-value < 0.05) was deemed significant.

## 5. Conclusions

In this study, anti-ZIKV_E IgY antibodies were induced in chicken in response for immunization of ZIKV_E recombinant proteins. The recombinant ZIKV_E proteins can be a potential target for vaccine development or developed as a point of care diagnostic tool, as demonstrated in western blot and ELISA assays. The anti-ZIKV_E scFv antibodies were selected from chicken-derived antibody libraries. These scFvs can be potential tools in inhibition and detection assays, as demonstrated in the figures above.

## Figures and Tables

**Figure 1 ijms-21-00492-f001:**
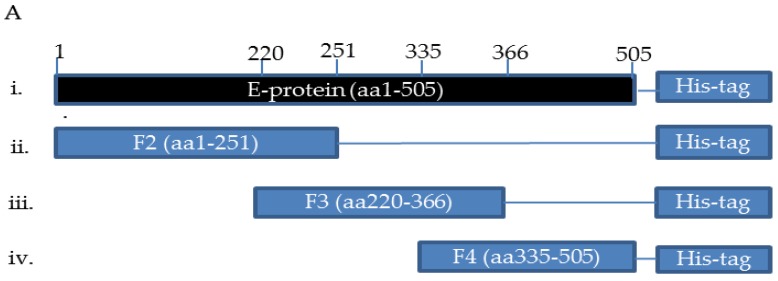
Purification, immunization, and characterization of ZIKV_E recombinant proteins. (**A**) Sketch diagram of plasmid constructs expressing His-fused ZIKV_E recombinant proteins. Three fragments are displayed, including full-length (ZIKV-E-protein); however, full-length and fragment F4 can be expressed yet difficult to purify (data not shown). (**B**) SDS-PAGE containing protein marker, soluble ZIKV_E F2 protein (27 kDa) purified from BL21 *Escherichia coli*, purified ZIKV_E F3 (17 kDa) protein from BL21 *E. coli*, and bovine serum albumin (BSA) as negative control. (**C**) Western blot detected with mouse anti-His tag antibody followed by rabbit anti-mouse immunoglobulin gamma (IgG) horseradish peroxidase (HRP)-conjugated antibody to confirm His-fused protein expression. (**D**) Immunization and egg collection schedule for humoral immune response titration.

**Figure 2 ijms-21-00492-f002:**
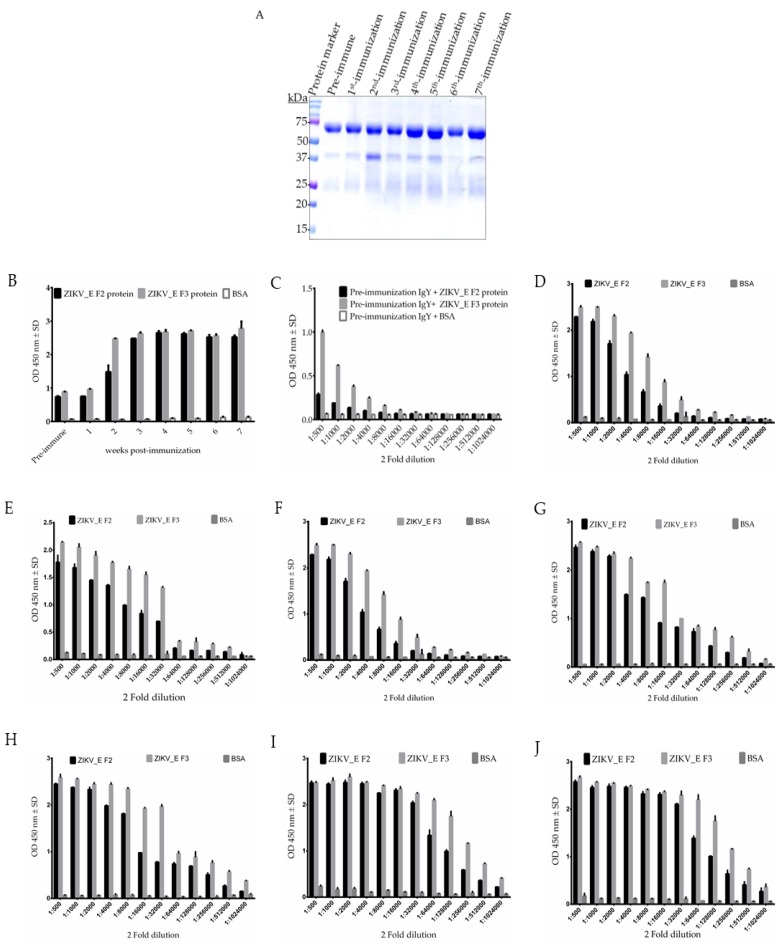
Characterization of purified immunoglobulin yolk (IgY) against ZIKV_E recombinant proteins. (**A**) Sodium dodecyl-polyacrylamide gel electrophoresis of a purified IgY antibodies. (**B**) The binding ability of purified ZIKV_E immunoglobulin yolk (IgY) antibodies from pre-immunization through seventh immunization on weekly basis. Immune response of (**C**) pre-immunization, (**D**) first immunization, (**E**) second immunization, (**F**) third immunization, (**G**) fourth immunization, (**H**) fifth immunization, (**I**) sixth immunization, and (**J**) seventh immunization was performed to assess the induced immune response before and after the final immunization.

**Figure 3 ijms-21-00492-f003:**
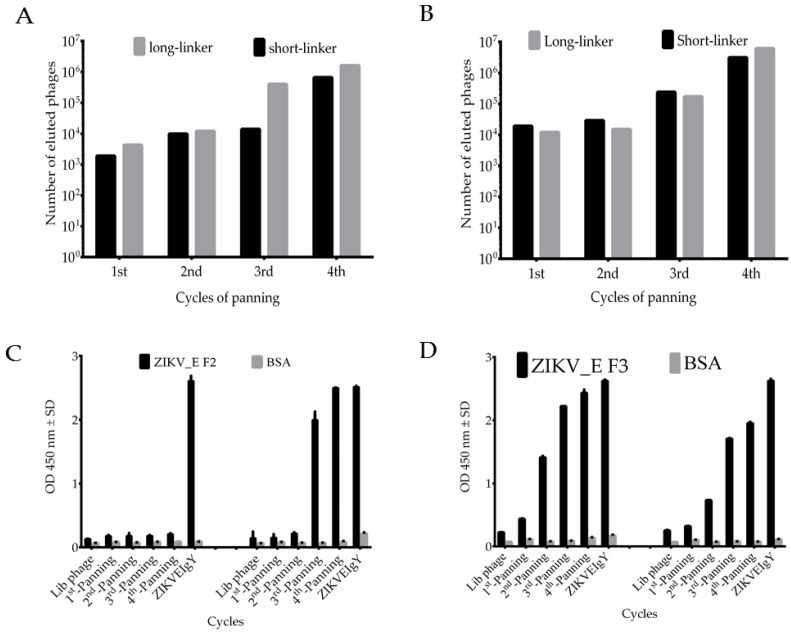
Selection and characterization of phage displayed anti-ZIKV_E single-chain fragment (scFv) library against ZIKV_E recombinant proteins. The biopanning procedure used constructed antibody libraries with short- and long-linkers. The eluted phage titers were determined after each cycle of panning for four rounds. (**A**) Number of eluted phages in terms of colony-forming units (cfu), (**B**) amplified phages tested against ZIKV_E F2 protein for each round of panning, (**C**) eluted phages obtained after panning against ZIKV_E F3 protein, and (**D**) amplified phages from each cycle of panning to test binding ability against ZIKV_E F3 protein. Anti-ZIKV_E IgY antibodies extracted from immunized chicken eggs were used as a positive control. The library phages were used as a reference point to the amplified phages of four round biopanning.

**Figure 4 ijms-21-00492-f004:**
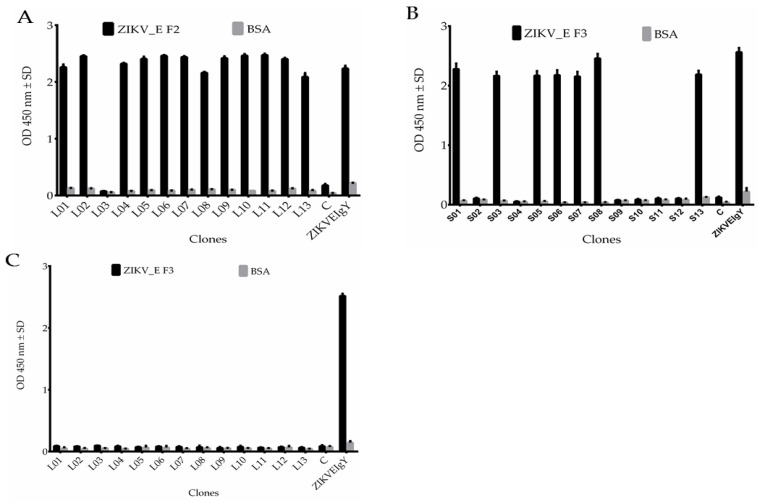
Sequence analysis, expression, purification, and characterization of four anti-ZIKV_E scFv antibodies. (**A**) Randomly selected scFv clones with long-linker against ZIKV_E F2 protein, (**B**) scFv clones with short-linker against ZIKV_E F3 proteins, (**C**) scFv clones with long-linker against ZIKV_E F3 protein. (**D**) The anti-ZIKV_E scFv antibodies with long- or short-linkers and their predicted amino acid sequences were compared with that of the chicken germ line. Framework region (FR) and complementary determining region (CDR) boundaries were indicated above the germ line gene sequences. Four representative clones were identified. (**E**) The scFv antibodies expressed in *Escherichia coli* (*E. coli*) were purified by nickel beads and analyzed on SDS-PAGE. (**F**) The identities of purified scFv antibodies were confirmed by western blot.

**Figure 5 ijms-21-00492-f005:**
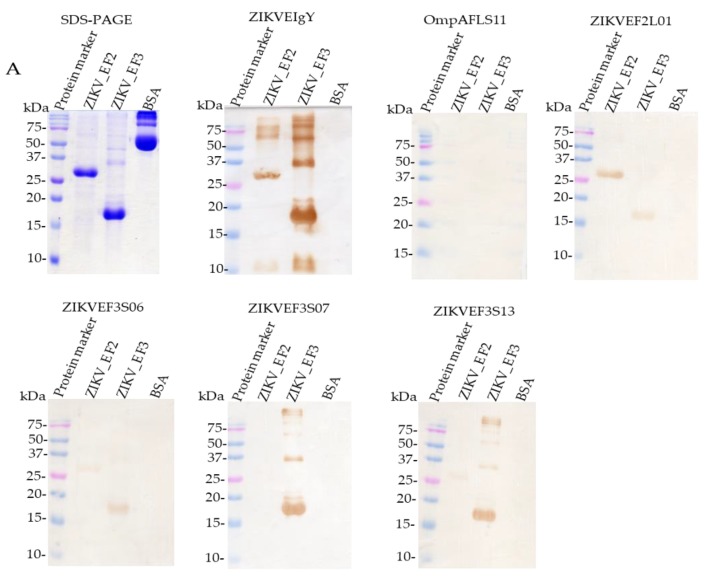
Characterization of binding specificity of anti-ZIKV_E scFv and polyclonal anti-ZIKV_E IgY antibodies. (**A**) Sodium dodecyl sulfate-polyacrylamide gel electrophoresis stained with Coomassie brilliant blue dye—lane 1: protein molecular weight marker, lane 2: ZIKV_E F2 protein, lane 3: ZIKV_E F3 protein, lane 4: BSA. The purified anti-ZIKV_E scFv antibodies (ZIKVEF2L01, ZIKVEF3S06, ZIKVEF3S07, and ZIKVEF3S13) were measured for their binding activities against two ZIKV_E recombinant proteins transferred on polyvinylidene difluoride (PVDF) membranes. Lanes 2: ZIKV_E F2 protein (27 kDa), lanes 3: ZIKV_E F3 protein (17 kDa). The bound scFv antibodies were detected using mouse anti-human influenza virus hemagglutinin (HA) tag antibodies (5000× dilution), followed by rabbit anti-mouse IgG HRP-conjugated antibodies (10,000× dilution). Anti-ZIKV_E IgY and anti-OmpAFLS11 scFv were used as positive and negative controls, respectively. (**B**) The purified scFv antibodies were analyzed for their ability to bind to ZIKV_E recombinant proteins immobilized on ELISA wells; the bars denote ZIKV_E F2 protein, ZIKV_E F3 protein, and bovine serum albumin (BSA). The assay was probed with ZIKVEF2L01 scFv, ZIKVEF3S06 scFv, ZIKVEF3S07 scFv, ZIKVEF3S13 scFv, ZIKVEIgY antibody (seventh immunization) as positive control, and OmpAFLS11 scFv as negative control. The scFv antibodies showed successful binding to antigens as indicated in the figures. (**C**) Flow cytometry analysis using the same antibodies plus 4G2 anti-flavivirus group antibody excluding anti-OmpAFLS11 scFv antibody.

**Figure 6 ijms-21-00492-f006:**
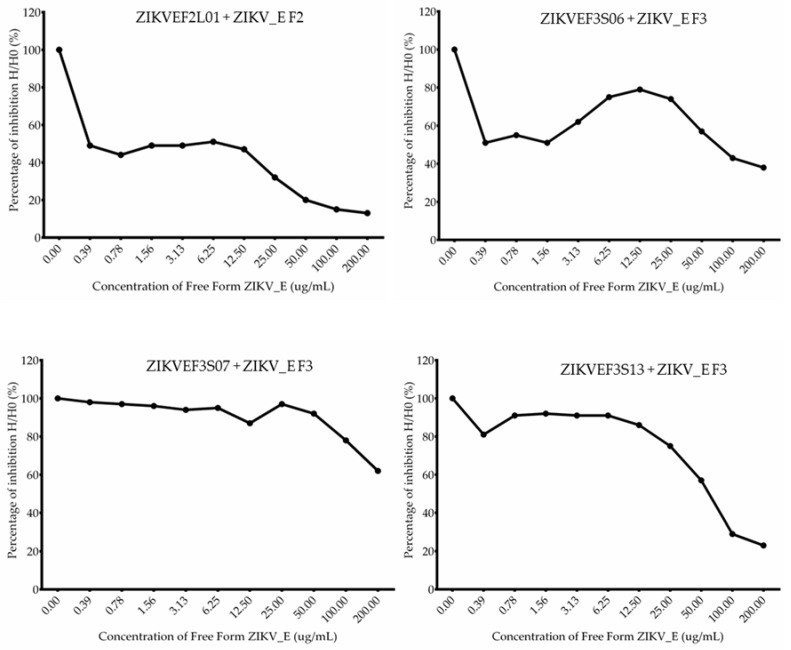
Competitive inhibition assay of anti-ZIKV_E scFv antibodies against ZIKV_E recombinant proteins. Purified scFv antibodies were first incubated with twofold diluted ZIKV_E recombinant proteins (200–0 µg/mL) and then loaded onto the ELISA plate coated with ZIKV_E recombinant proteins. The bound scFv antibodies in the presence or absence of soluble ZIKV_E recombinant proteins are presented as H/H0 to show the inhibitory percentage of the binding specificity. The results are presented in triplicate.

**Table 1 ijms-21-00492-t001:** List of primers used for PCR amplification of *Zika virus* envelope (ZIKV_E) gene from pcDNA3.1_ZIKV E plasmid DNA.

Primer Name	Oligonucleotide (5′–3′)
ZIKVEFLF	TAAAATGGATCCATGATCAGGTGCATAGGAGTCAGC
ZIKVEFLR	AATTTTCTCGAGAGCAGAGACGGCTGTGGATAAG
ZIKVEF3F	AATTAAGGATCCGACATTCCATTACCTTGGCACGCT
ZIKVEF3R	TTAATTCTCGAGGATTACGGGGTTAGCGGTTATCAAC
ZIKVEF2R	TAAATTCTCGAGGGCATGTGCGTCCTTGAACTCTAC
ZIKVEF4F	TTATAAGGATCCACAGATGGACCTTGCAAGGTTCCA

Underlined nucleotides: digested sites for restriction endonucleases. ATG is underline to show that it was added to the original sequence. It could not be digested during PCR process. The start codon was added because the ZIKV envelope gene sequence has no ATG codon from the beginning of the sequence (Appendix A).

**Table 2 ijms-21-00492-t002:** Amino acid mutation rates of the light and heavy chain genes of the anti-ZIKV_E scFv antibodies.

Region	FR1	FR2	FR3	FR4	Total FRs	CDR1	CDR2	CDR3	Total CDRs
VL	5–10%	6–19%	9–16%	0–9%	5–19%	50–88%	29–71%	36–64%	29–88%
VH	10–15%	0–7%	9–19%	0%	0–19%	20–100%	29–47%	63%	20–100%

FRs: Framework regions; CDRs: complementarity determining regions; VL: light chain of the variable region; VH: heavy chain of the variable region.

**Table 3 ijms-21-00492-t003:** The calculated dissociation constant (KD) values of anti-ZIKV_E single chain variable fragment (scFv) antibodies.

scFv Clones	Inhibition of 50% Binding of Total Protein (µg/mL)	*K_D_* Values (M)
ZIKVEF2L01	0.5515	2.04 × 10^−8^
ZIKVEF3S06	77.425	4.55 × 10^−6^
ZIKVEF3S07	254.14	1.49 × 10^−5^
ZIKVEF3S13	110.885	6.52 × 10^−6^

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
