# Peer review of "Expression, Purification, and Characterization of Anti-Zika virus Envelope Protein: Polyclonal and Chicken-Derived Single Chain Variable Fragment Antibodies"

_ijms, 2020, doi:10.3390/ijms21020492_

Round 1
Reviewer 1 Report
The authors have greatly improved their manuscript; They have carefully considered all my previous suggestions and have implemented them to the best of their abilities. Thus, the manuscript is now suitable for publication in IJMS.
Reviewer 2 Report
The authors made significant changes to the manuscript and answer all the concerns that I have regarding this manuscript. A more detail description of methods and figures was edited in the new manuscript. Those changes would help people who interested to reproduce the conclusion. I highly recommend publishing this manuscript.
This manuscript is a resubmission of an earlier submission. The following is a list of the peer review reports and author responses from that submission.
Round 1
Reviewer 1 Report
The authors developed single-chain variable fragment (scFv) antibodies using phage display technology and characterized three candidates and the manuscript was very well written. I have a few minor comments.
Figure 1C, the WB results showed the purification of ZIKV_E F2 and F3 seems to have lots of unspecific bands for ZIKV protein but can be detected by WB. I am just wondering why it would happen? Figure 3B and D, the color is not very corresponding to the Figure, such as 3rd spanning and 4th spanning, it should fill with black? Figure 4, if look at the WB, it seems that scFV forms a dimer? would that be possible?Author Response
The responses to reviewer 1 comments have been attached.

Reviewer 2 Report
Peer-review report for the International Journal of Molecular Sciences
Article: Expression, purification, and characterization of anti-Zika virus envelope protein: polyclonal and chicken-derived single chain variable fragment antibodies
Overview: In this report, Fellow Mwale et al. explained how they have developed single chain variable fragment (scFv) antibodies that target the ZIKV envelope protein using phage display technology. They show that they successfully immunized chickens using their recombinant antigen; That they were successful to construct a cDNA library from the immunized chickens to support biopanning of phage displayed peptide libraries. Lastly, they show that successive round of biopanning successfully isolated anti-ZIKV_E scFv antibodies of increasing affinity towards their intended ligand.
Introduction
General comments:
Introduction needs to be rewritten to follow a “logical flow” of ideas, avoid misleading generalizations. The author needs to provide the reader with enough information to understand where their research fit in their specific niche of ZIKV biology. Bibliography should be revised to include proper references. Specific comments are provided below to guide the authors.
Specific comments:
Line 43: Zika virus is indeed transmitted by Aedes mosquitoes. However, other modes of transmission has been described: please consider rephrasing this statement to include: vertical transmission from mother to fetus and direct blood/biological fluids contact via blood transfusion or sexual contact.
Line 44: please consider changing “Zika has been attributed” to “Zika has been associated” in this sentence. Moreover, please re-phase to contrast comorbidities that occurs in pregnant women/fetus and healthy adults (ex. Zika-virus microcephaly does not occur over the course of infection in healthy adults.)
Line 49. Please consider including this reference (Mlakar, J. et al. Zika Virus Associated with Microcephaly. N. Engl. J. Med. 374, 951–958 (2016).) in addition, or instead of Satterfield-Nash et al. 2017. as the latter paper is much more a review of evidence and not an original citation.
Line 49: “Study also suggested that individuals with compromised immunity could be more susceptible to ZIKV infection and disease development” => This statement is not supported by the Yockey et al. report provided in reference that studies Zika infection in various mouse models with genetically altered immune systems.
Line 55 to 61: Please consider moving it to line 43 for a better flow of ideas: this section fits with the description of the virus.
Line 59: Reference 18 is a general review of Hepacivirus and Flavivirus and does not specifically support the statement made in 57 to 59; Please update citation to include a review paper that specifically talk about ZIKV genomic organization and life cycle.
Line 59: Reference 19 (Virion assembly and release. Curr top microbiol immunol 2013) is a Hepatitis C paper; please change to review paper more appropriate to the topic.
Line 61: ZIKV infection is generally misdiagnosed because its signs and symptoms overlap with other endemic viral infection such as Dengue and Chikungunya. Thus diagnosis, of a symptomatic ZIKV infection, relies heavily on serological evidence rather than clinical presentation. Please consider rephrasing.
As a suggestion, I would move line 51 to 54 after this paragraph to tie your introduction text to your perspective/unmet medical need and then introduce your solution: scFV.
Line 61: Reference 14 (Swanstrom et al., 2014) is not appropriate to support clinical description of ZIKV infection as it is basic science paper investigating neutralizing human monoclonal antibodies that naturally occurs during ZIKV infection in humans: please change accordingly.
Line 63: reference 14 is out of order: it appears in the text after reference 21; please update bibliography accordingly.
Line 64: please rephrase in context and add a reference: passive immunization using phage display technologies have been described. It would properly introduce the novelty of your method to produce scFV in chicken vs. what has been done before.
Line 73: Please add reference to your strain isolate (genebank # KU497555) or cite original paper for Brazil-ZKV2015 : Calvet, G. et al. Detection and sequencing of Zika virus from amniotic fluid of fetuses with microcephaly in Brazil: a case study. Lancet Infect. Dis. 16, 653–660 (2016).
Results
Section 2.1
General comments:
Please revise the Figure according to the specific comments below. The Figure is a bit confusing since the labels used for the sub-panels are reused (ie. 1-2-3, etc.) but are not depicting the same samples. Please label the lanes so that the reader does not need to read the Figure legend to understand your results.
Specific comments
Table 1.
Please check the primer sequence provided. I was not able to match any of the sequence provided to a known viral nucleotide sequence or any nucleotide sequence of NCBI’s BLASTn suite. In your response to the reviewer’s comments, please provide the alignment of your primer to the ZIKV reference sequence and strain specific sequence and provide agarose gel of the amplified PCR products. In addition, please revise the table to underline the nucleotides added for the introduction of your restriction sites.
Figure 1.
Line 84 to 86: Figure 1B only shows the expression of F2 and F3. Please rephrase the text according to the data as not all of the proteins of Figure 1A were successfully expressed. Also, please state detection method for subpanel B: ie. Coomassie blue, silver stain, western blot.
Line 89-90: please re-phase to explicitly described the immunization schedule ( ex. Chicken received an intramuscular injection of 50 ug of a mixture of antigens each 7 days for 56 days .). Please re-phase the method section as there are 2 regimens mentioned and it is not clear to me if they were immunized with 50 ug or 300 ug IM q7D.
Line 94-95: “IgY antibodies were digested by beta-mercaptoethanol” => it is not clear to me what you want to say by digested.
Line 95: “The concentration of purified IgY antibody was estimated to be ~82.7 mg/mL using BCA protein quantification assay.” Please be more specific. Was it 82.7 at reach time point, a mean, etc. Re-phase accordingly.
Line 96: “Taken together, the results demonstrated that the immune response through IgY antibodies analysis were successfully elicited in chicken.” => conclusion is not supported; Figure 1E does not show a successful immunization: IgY are present in the pre-immune sample and does not drastically change between time points. Data from section 2.2 are needed to show that a specific immune response was elicited. Please rearrange the text accordingly.
Section 2.2
General comments:
Please revise the Figure according to the specific comments below. Some key elements are missing from Figure 2:
What are the error bars? Are they Standard Deviation, Standard Error of the Mean, etc. Please add a description to the Y-axis label “OD 450 nm ± SD” You are using the word “significant” in the description of the results; yet, there are no statistics provided. Please rephrase the text to indicate a trend or include an appropriate statistical test to support your choice of words. The rational to show the data for pre-immune and 7th immunization only is not explicit; Please include all the ELISA data, from each time point in the Figure or as a supplementary, or explain why the assay is not shown or was not done for the first, second, third, fourth, fifth and sixth immunization. Please change the figure format to a bar graph with individual data points to show the variance and distribution of each dataset.Specific comments:
Figure 2
Line 124: “This suggests that the elicited immune response in chickens were increased after the first shot (1st immunization), then steadily increased until reaching a plateau with the 3rd-immunization. “ => Please rephrase to give a better description of the results (ex. The plateau seems to be reached after the 2nd immunization for the F3 antigen.)
Line 132: “Together, the data suggest that the immune response was effectively raised after 3rd-immunization.” => please rephase; according to your response to my previous comments.
Section 2.3
General comments:
I believe that Figure 3 is contrasting the two antibody libraries made from F2 and F3 in subpanel A/B and C/D, respectively. This is not clear from reading the Figure, Figure legend or text. Please rephase the whole section to describe the results with clarity and avoid confusion.
Please revise the Figure according to the specific comments below. Some key elements are also missing from Figure 3:
What are the error bars? Are they Standard Deviation, Standard Error of the Mean, etc.? Please add a description to the Y-axis label “OD 450 nm ± SD” There are no statistics provided. If possible, please include an appropriate statistical test to support your findings. Please change the figure format to a bar graph with individual data points to show the variance and distribution of each dataset.Specific comments:
Line 156: Your previous paper, provided in reference, does not show any data that support that claim. The only data related to panning seems to be Figure S2 which shows a baseline OD of about 0.8 an increase to a little bit over 1 after the 3rd panning. Please re-phase accordingly.
Section 2.4
General comments:
In order to support the claim of this section, you need to show, experimentally, that the selected clones have a higher affinity or an affinity improvement that correlates with their increase “burden” in mutation. Please determine the binding EC50s of each individual, or pool, of clones to show that they have indeed, a higher affinity. Otherwise, please re-phase.
Additionally, I am not quite sure of the added value of this section. Please consider a rewrite with the data from section 2.5.
Specific comments:
Line 181-184: Please re-phase. I am not 100% sure of the message that you are trying to convey. Also, please make a logical link to the reference provided. This could be done by separating your toughs into more than one sentence.
Line 197: “four representative clones were identified”. => they are not representative if you have only isolated four clones; please rephrase.
Section 2.5
General comments:
Please revise the Figure according to the specific comments below. The Figure is a bit confusing since the labels used for the sub-panels are reused (ie. 1-2-3, etc.) but are not depicting the same samples. Please label the lanes so that the reader does not need to read the Figure legend to understand your results. This is especially true for Figure 5B. Below is an example of what your Figure5B should look like.
Specific comments:
Line 209-210: “All scFvs (ZIKVEF2L01, ZIKVEF3S06, ZIKVEF3S07, and ZIKVEF3S13) recognized ZIKV_E truncated proteins with different binding activity” => There are many factors that come into play during “western blotting” that could indirectly alter the binding activity of an antibody. Thus, I am not comfortable with the authors premise that a lower signal from a western blot is a conclusive evidence of an antibody lower affinity. The authors own Figure 5B shows this interpretation (see clone LO1). I would suggest that the authors re-phase to moderate their statement based on the data available in Figure 5A.
Line 221: “Taken together, this implies that the significant immune response was induced in chickens.” => please rephrase according to the data.
Section 2.6
General comments:
In this section, the authors provide the results of a competitive ELISA assay for each of the four clones previously selected. To provide an adequate evaluation of the results, the authors need to improve the following aspect of this experiment:
Provide a cross-reactivity analysis that shows that competitive ELISA is not affected by the presence of unrelated or somewhat related antigens. Refine their assay to get a better sensitivity as their competitive ELISA should follow a sigmoidal curve; I recommend running a checkerboard titration to find the optimal concentration to be used in this assay. Review the Figure; it is difficult to understand how the percentage of inhibition decrease with an increasing amount of Free Form antigen; Also, from what I can gather from the scFv literature; competitive ELISA in generally done using ng/mL dilution of competitors. Here, the label shows ug/mL, which would indicate a relatively low-binding affinity.

Author Response
The responses to reviewer 2 comments have been uploaded.
